# Healthcare Services and Formal Caregiver’s Psychosocial Risk Factors: An Observational Study

**DOI:** 10.3390/ijerph19095009

**Published:** 2022-04-20

**Authors:** Anabela Pereira, Elisabeth Brito, Isabel Souto, Bruno Alves

**Affiliations:** 1Research Centre on Didactics and Technology in the Education of Trainers (CIDTFF), University of Aveiro, 3810-193 Aveiro, Portugal; 2Department of Education and Psychology, University of Aveiro, 3810-193 Aveiro, Portugal; isabel.souto@ua.pt; 3Research Unit on Governance, Competitiveness and Public Policies (GOVCOPP), University of Aveiro, 3810-193 Aveiro, Portugal; ebrito@ua.pt; 4School of Technology and Management of Águeda, University of Aveiro, 3810-193 Aveiro, Portugal; 5Jean Piaget Higher Institute, 4405-678 Vila Nova de Gaia, Portugal; pcjbta@gmail.com

**Keywords:** formal caregivers, healthcare services, psychosocial risk factors, work-related stress, chronic diseases

## Abstract

The prevention and management of chronic disease primarily requires risk reduction measures, through strategic coordination across various government areas. Recognizing that health workers and the public health workforce are integral to building strong and resilient health, the present study analyses the relation between Psychosocial Risk Factors (PRFs, to which formal caregivers are exposed in the healthcare settings), and the work system related elements of the Systems Engineering Initiative for Patient Safety (SEIPS 3.0) framework. An empirical study was carried out, involving 333 formal caregivers of healthcare services. A total of 31 PRFs were assessed (using the COPSOQ III), making it possible to find a relationship between the PRFs analyzed with three elements of the work system, namely Task (5 PRFs), Organizational factors (17 PRFs), and Individual (9 PRFs). The present work contributes not only in terms of outcomes that allow the development of mental illness prevention and mental health promotion actions for healthcare formal caregivers, but also in terms of the relevance that these factors can have on the quality of health services, as well as their users (patients), in line with SEIPS 3.0 model.

## 1. Introduction

The Sustainable Development Goals (SDG) 2030 Agenda, established by the United Nations, present a set of challenges that ought to be considered by healthcare services. Crucially, “Ensure healthy lives and promote well-being for all at all ages” is the main goal that SDG3 advocates, which includes as a target (among others), “reduce by one-third premature mortality from non-communicable diseases (NCDs) through prevention and treatment and promote mental health and well-being”. According to the World Health Organization (WHO), NCDs (also known as chronic diseases), not only account for 71% of all global deaths, but also have a huge impact for individuals, families and communities, and threaten to overwhelm health systems [1,2]. The prevention and management of chronic disease primarily requires risk reduction measures, through strategic coordination across various government areas (health, finance, transport, education, planning, etc.) [1,2,3]. Even so, it is important to recognize that NCDs, represent a particular challenge to health systems at their different levels (primary, secondary and tertiary). Clearly, the demands on healthcare services and professionals go beyond the needs that arises from preventive action but include an effective response to the sudden manifestations associated with long-term illness. The focus is on the ability of health systems to respond more effectively and equitably to the healthcare needs of people with NCDs, by transforming and strengthening these systems to provide more effective, efficient and timely care [2,3]. On the other hand, NCDs as diseases of prolonged evolution/late-latency pose the challenge that in order to improve the quality of healthcare provided, it is also necessary to understand patient journey in healthcare [4,5]. In this sense, the Systems Engineering Initiative for Patient Safety model (SEIPS 3.0) stands out [4]. This systemic approach refers to the series of interactions of patient with healthcare (over space and time), in a multi-level system of the patient journey, over time, in healthcare, that includes not only the work system (local systems where patients interact with health professionals in a given time and space), the socio-organizational context (which may include formal and informal health care network), and the external environment of the healthcare organizations [4]. As any system, it is subject to continuous changes and adaptations that can be influenced by the external environment, as well as through learning and improvement mechanisms, arising from feedback. More specifically, SEIPS 3.0 integrates the complexity of interactions between the local context or system of work, the socio-organisational context, and the environment external to the health system itself.

With specific regard to work systems, the SEIPS 3.0 model includes five key elements, namely: (1) Environment (specifically ergonomic factors); (2) Task (e.g., perceived quantitative workload, work pressure, cognitive demands, job content); (3) Technology (lack of adequate skills to use the technology, fear over job loss due to replacement by technology, physical characteristics of the tools and technology/poor workstation design); (4) Organizational factors (e.g., workload and rate, work schedules, work related decisions participation); (5) Individual (e.g., cognitive and psychosocial dimensions, physical health status, skills and abilities, physical conditioning, anthropometrics, prior experiences and learning, motives, goals and needs) [4,5,6]. These elements interlink to provide the strengths and resources for the realization of individual and organizational goals. However, on the other hand, they also comprise a set of factors that may also contribute negatively in terms of motivation, performance and work-related distress (Occupational Stress—OS), namely, several Psychosocial Risk Factors (PRFs) [6]. We clarify that PRFs can have origin in either content or context of work. Content factors include several categories that fall into Task element of work systems, as well as contextual PRs factors that may come from Organizational factors. Individual factors can also interfere with this dynamic [6,7,8,9].

The present work focuses on these work system-related elements, specifically in the consideration of PRFs related to them. More specifically, this study aims to analyze the PRFs (and respective health impact) to which formal caregivers are exposed in healthcare settings, in an integrated way with the work system-related elements of the SEIPS 3.0 framework (Environment, Task, Technology, Organizational and Individual factors). We intend to draw attention to the relevance of these factors can have on the quality of health services, as well as their users (patients), in line with SEIPS 3.0 model. Cumulatively, the results will alert for the development of mental health promotion actions based on scientific evidence for healthcare formal caregivers.

## 2. Materials and Methods

### 2.1. Study Design

The present work integrates an observational study conducted between April and August 2021. The aim was to characterize the work context of healthcare formal caregivers’, through a quantitative methodology, using an online questionnaire survey. In this paper, the first results of the study are disclosed, specifically the data on exposure to PRFs (and respective health impact), to which formal caregivers are exposed in Portuguese healthcare settings.

### 2.2. Participants

The study relied on participation of 333 formal caregivers of Portuguese healthcare services. The sample was constituted based on the total target population of 88,174 health professionals working in Portugal [10]. Considering the error of 5% and the confidence level of 95% (1.96), the representative sample should be about 383 participants. However, the percentage of the sample by group follows a distribution, in general, similar to that found in the population percentage, in terms of gender, region or public/private sector.

### 2.3. Instruments

Besides the collection of demographic and work characteristics of participants, the instrument Copenhagen Psychosocial Questionnaire (COPSOQ III, medium version) was employed.

The COPSOQ is a highly reliable scale for assessing PRFs in the workplace. It is a powerful tool that gathers international consensus on the adequacy for evaluating an extensive and most important psychosocial dimensions [9,11,12,13,14]. The COPSOQ differs from other scales, since it systematically approaches the interaction between psychosocial work environment and health, not being based and limited to a specific theoretical model, providing a versatility of uses [9,13]. Meanwhile the International COPSOQ Network released an updated instrument [14,15], the COPSOQ III, whose middle version also has a Portuguese version [11]. This revised instrument is composed by 85 items distributed in 31 dimensions, namely: Quantitative demands, Work pace, Cognitive demands, Emotional demands, Influence on Work, Development possibilities, Control over working time, Meaning of work, Commitment to workplace, Predictability, Recognition, Role clarity, Role conflicts, Quality of leadership, Social support from colleagues, Social support from supervisors, Sense of community at work, Job insecurity, Insecurity over working conditions, Quality of work, Horizontal trust, Vertical trust, Organizational justice, Work–life conflict, Job satisfaction, Self-rated health, Self-efficacy, Sleeping troubles, Burnout, Stress, and Depressive symptoms. The Portuguese COPSOQ III (middle version) show good internal consistency. In the present study, also show good internal consistency with a range between α = 0.665 and α = 0.939 for the subscales.

A sociodemographic questionnaire developed by researchers, aimed to collet demographic characteristics (e.g., age, gender, marital status), as well as information and specificities of the work (e.g., healthcare sector, work geographic area).

### 2.4. Procedures

To recruit participants, the research protocol was disseminated by email, through the communication services of the target entities (healthcare services), as well as disseminated through social networks. Participants were invited to complete the online questionnaires on a voluntary, anonymous, and confidential basis, adopting a natural random selection of the sample. Several dissemination reinforcements were made in an attempt to obtain sample representativeness based on target population data and proportional criteria.

All data collection, registration and communication procedures were in accordance with the guidelines of the National Commission for Data Protection and the General Data Protection Regulation. This implied the observance of prior informed consent, given voluntarily and in an informed manner, based on adequate and sufficient information, namely, objective, purpose and procedures in data processing. All information was treated with the highest degree of confidentiality. The present work also considered the Code of Responsible Conduct in Scientific Research, subscribing to the principles, rules and procedures of the European Code of Conduct for Scientific Integrity [16].

### 2.5. Data Analysis

To calculate the health impact that exposure to PRFs represents, the average obtained in each factor of COPSOQ was placed in a division of tripartite percentiles, with respective cut-off points of 2.33 and 3.66 (according to the alternative mode of assessment of COPSOQ II) [13]. Therefore, each factor can be interpreted by means of the health impact that the exposure represents, in particular, a health-friendly situation, an intermediate health situation and health risk.

To group the PRFs (assessed with COPSOQ III) into the elements of the work system of the SEIPS 3.0 model, a content analysis was performed by three independent reviewers (psychologists), using as a reference the description of the five elements of the work system (Environment, Task, Technology, Organizational factors and Individual) [6]. Disagreements between the evaluators were resolved through discussions or consultation with the research team. An effort was made to maintain the reliability of the content analysis, and the need for a concordance of at least 90% was determined [17].

## 3. Results

### 3.1. Sample’s Sociodemographic Characteristics

The sample was composed of 333 formal caregivers of public (*n* = 287) and private (*n* = 46) healthcare services. It comprised 252 females (75.7%) and 81 males (24.3%), aged between 22 and 67 years old (*M* = 43.52, *SD* = 10.121). The participants had different professional roles: Nurses (*n* = 211), Doctors (*n* = 47), Higher Technician in Diagnosis and Therapy (HTDT, *n* = 36), Technical Assistant (*n* = 19), Healthcare assistant (*n* = 12) and other healthcare-related professionals (*n* = 8). Regarding regional location, the sample is from several Portuguese geographic regions, namely North (41.1%), Center (44.7%), Lisbon Metropolitan Area (10.5%), Alentejo (0.6%), Algarve (0.6%), Madeira Autonomous Region (0.3%), Açores Autonomous Region (0.6%). There was also 1.2% of the sample that worked in more than one region, and 0.3% did not provide region information.

### 3.2. Psychosocial Risk Factors Exposure

Considering the outcomes of the analysis of the PRFs (through COPSOQ), it was possible to identify a relationship with three of the five elements of the work system of SEIPS 3.0 model, namely Task (PRFs associated with work content), Organizational factors (PRFs associated with work context), and Individual (PRFs associated with individual work-related behaviors, as well as health quadrant).

In the element of Task, it was possible to frame five PRFs (Figure 1). It is worth pointing out that out of these PRFs, only one showed high percentages of a health-friendly situation (a favorable situation in terms of the impact that exposure represents), namely, Quality of Work (54.4%). On the other hand, in the remaining four, the higher percentage of answers correspond to high risk for health or an intermediate situation. The highest percentage of high health risk responses is found in exposure to the PRFs of Cognitive demands (78.1%), Emotional demands (73.6%), and Work pace (48.0%). Quantitative demands show a higher percentage of intermediate health risk, in exposure to the PRFs represents (57.1%).

In the element of Organizational factors, it was possible to frame 17 PRFs (Figure 2). Out of these PRFs, only five showed high percentages of a health-friendly situation (in terms of the impact that exposure represents), namely, Role clarity (75.7%), Development possibilities (75.4%), Sense of community at work (60.7%), Job insecurity (57.1%), and Social support from colleagues (53.2%). On the other hand, only in one did a higher percentage of answers correspond to high risk, namely Work–life conflict (51.1%). The remaining ten showed a higher percentage of intermediate health risk, in exposure to the PRFs represents (Figure 2). However, it is worth pointing out that the sum values of responses framed in a high health risk or intermediate health situation of the PRFs, Predictability, Role conflicts, Social support from supervisors, Work–life conflicts, and Organizational justice, were above 75%.

Finally, in the element Individual, it was possible to frame nine PRFs (Figure 3). Out of these PRFs, two showed high percentages of a health-friendly situation, namely, Meaning of work (81.1%), and Commitment to workplace (48.6%). Self-efficacy and Job satisfaction show a higher percentage of an intermediate health risk, in exposure to the PRFs (57.4% and 48.6%, respectively). All the remaining PRFs correspond to health dimension, which shows the most worrying results: a higher percentage of answers correspond to high risk for health or an intermediate situation, whose sum percentages of responses are mostly above 75% (Self-rated health, Burnout, Stress and Depressive symptoms), or above 50% (Sleeping problems).

## 4. Discussion

The present study aims to look more specifically at the PRFs (and their impact on health) to which formal caregivers are exposed in workplaces, in an integrated way within the SEIPS 3.0 framework related to the work system (Environment, Task, Technology, Organizational and Individual Factors). A total of 31 PRFs were assessed (using the COPSOQ III), making it possible to find a relationship between the PRFs analyzed with three (of the five) elements of the work system, namely Task (5 PRFs), Organizational factors (17 PRFs), and Individual (9 PRFs). The main results show that the Task and Individual elements present the most worrying results, as they comprised PRFs with higher rates for health impact.

In the Task element, 2 PRFs shows response percentages above 75% in high risk for health (Cognitive demands, Emotional demands), also corresponding to the highest values of all assessed PRFs. At the level of the Organizational factors’ element, it frames most of the PRFs that had a high percentage of answers corresponding to a friendly health situation. Only the PFRs of Work–life conflict showed a high percentage of answers corresponding to a high risk to health. The remaining ten PFRs presented a high percentage of answers corresponding to immediate-term health risk. However, summed values of responses framed in high and intermediate health situation of the PRFs, Predictability, Role conflicts, Social support from supervisors, Work–life conflicts, and Organizational justice, are above 75%. In turn, in the Individual element, the domains of health stand out (Self-rated health, Burnout, Stress, Depressive symptoms, and Sleeping problems), whose percentage of responses in high health risk and intermediate health risk summed, are above 75%.

It should be recalled that the elements of Task and Organizational (of SEIPS 3.0 work system), include work content and context factors, while the Individual element includes individual characteristics of workers, particularly in the health quadrants. As such, the present results draw attention, first and foremost, to the need to reinforce the central role and responsibility of organizations in the occupational safety and health (OSH) of their workers [18], particularly healthcare caregivers. It should be noted that some European countries include workplace components in their health programs; however, mental health policies (and respective action plans), are the least represented in the policy framework [19], which worsens substantially if we focus attention on work-related mental health policies at work. In Portugal, the assessment and prevention of PRFs in the work context is covered in the legislation on OSH (Law No. 3/2014 of 28 January 2014). As stated by Jain et al. (2018), “*Recognition, prevention and treatment of both occupational diseases and accidents, as well as the improvement of recording and notification systems are high priorities for improving the health of individuals and the societies they live in*” [18]. The present findings reinforce that PRFs management must go beyond legal assessment requirements and it is urgent that public policies recognize that assessing PRFs is just one part of the overall process. In fact, a risk assessment needs to be carried out prior to making an intervention [18], but is imperative to develop guidelines concerning the following: (1) the minimum prescriptions for exposure levels to PRFs that allow maximizing the prevention (similarly to what is foreseen in physical, chemical and biological risks management framework); (2) the inclusion of prevention for new and emerging risks, as well as focus on work-related diseases and not only on work-related injuries; (3) standards and tools that allow maximizing the intervention process of PRFs exposure’ consequences, including long-latency diseases, associated with work-related distress exposure.

Additionally, results from the Individual element reinforce that PRF-related actions must go beyond the primary level at the organizational level (prevention). It is also necessary to be alert to health services caregivers’ individual needs (early diagnosis and prompt treatment of health impact), otherwise there is a risk of encouraging a vicious circle in which the formal caregivers themselves will suffer impacts in both ways (as professionals and as patients). It is emphasized that workers exposed to high work-related distress may develop psychological morbidities (e.g., anxiety, depression, burnout), and/or develop physical health problems (e.g., musculoskeletal disorders) that may significantly affect the flow of work, through decreased work capacity, less dedication to work, low productivity and/or unsafe work practices, which may result in an increased accident rate, as well as a relation with NCD’s development [20,21,22,23,24,25,26,27,28,29,30,31,32,33,34,35]. There emerges a need to provide instruments that encourage the empowerment of health professionals. The relevance of the development of e-health solutions is highlighted as one of the priorities emerging from the pandemic phase, in several areas of action [36,37], which could be also recommended for occupational health.

On the other hand, it should be noted that there are relevant intervention approaches focused on the development of skills for stress management and anxiety reduction. Mindfulness training protocols has demonstrated some effectiveness, contributing to guide attention, support resilience, process emotion and also for good mental health, has gained increasing recognition and has shown evidence of its possible effectiveness [38,39]. Another approach is based on promoting better postures in the workplace, addressing work-related musculoskeletal disorders (MSD). A progressive sedentary lifestyle, associated with work-related musculoskeletal disorders, remains one of the most common work-related problems in the European Union (EU). A large majority of all EU workers report MSD-related complaints and stress, depression, and anxiety as the most serious health problems [22,31]. In addition, while it is true that OSH has recently become a much higher priority [18], particularly in PRF exposure and work-related distress in light of the growing evidence from interventional measures in workplaces, it is also true that intervention tools that are fit for purpose with work-related relevance need to be developed [32,40]. As such, this study contributes data on the specific needs of the professionals, which can help maximize prevention and intervention processes for formal caregivers in health services. Note that the OSH predisposing factors management paradigm has been recommended as more effective than simple workplace interventions; as such, there is an expectation to be more effective in workers’ health protection and promotion [24,32,41]. Although these results are focused on PRFs (and their respective health impacts), in an integrated way with SEIPS 3.0 (work system), they also allow the identification of opportunities for improvement in the patient’s journey through the health system. It is also reiterated that this should be a concern that should be rapidly integrated in the action plans in response to NCDs, recognizing that health workers and the public health workforce are integral to building strong and resilient health [2,3].

Considering the overall results, we realize that prevention and intervention actions are urgent, in two ways: (1) to reply to health services caregivers’ considering mainly Task and Individual needs; (2) to improve the patient’s journey through the health system (given relevance that these factors may have on the quality of health services as well as on their users) [4]. It is also important to mention that given the relationship between exposure to work-related stress and the development of NCDs [18,42], prevention and intervention actions can also have a positive impact on the prevalence of NCDs.

Studies on this topic contribute to the strategic plan of Horizon Europe 2021–2024 (health cluster), as well as to the United Nations agenda, in the above-mentioned SDG3, and the addition to SDG 8: “*protection of rights at work and promotion of safe and secure work environments, achieving full and productive employment and decent work for all women and men*”.

It should also be noted that the data was collected during the COVID-19 pandemic, via a survey conducted online; therefore, we must warn of the limitations inherent in this methodology. Whilst reflecting the particular reality and possible factors emerging from this phase is highly important, it is recognized that looking only at one system of work at one point in time is limiting and does not consider the temporal context of changes and adaptations taking place, as well as the navigation of patients between and among healthcare organizations [4]. For future studies, we propose focusing on a long-term investigation, which may help to create a more integrated understanding of the phenomena.

## 5. Conclusions

The present study analyses the relation between PRFs (to which formal caregivers are exposed in the healthcare settings), with the work-system-related elements of the SEIPS 3.0 framework. A total of 31 PRFs were assessed (using the COPSOQ III). Of the total PRFs, 7 showed high percentages of favorable situation responses (in terms of the impact that exposure represents), 19 PRFs presented high rate of responses in intermediate health situation, and 5 showed high percentage of responses in high risk for health. The relation between the PRFs analyzed with three elements of the work system (Task, Organizational factors, and Individual) were presented and discussed.

PRFs management must go beyond legal assessment requirements, and it is urgent that public policies recognize that assessing PRFs is just one part of the overall process; it is imperative that guidelines are developed for preventive effectiveness. Additionally, it is reinforced that PRF-related actions must go beyond the primary level (prevention), as it is necessary to also act at the secondary and tertiary levels (early diagnosis and prompt treatment of a disease).

The main findings confirm that there are some specific main domains that should be considered, not only in mental illness prevention but also in mental health promotion, as proposed by the European Agency for Safety and Health at Work [18,19,33]. In general, these results also confirm that prevention guidelines and intervention tools should be fit for purpose with work-context specific needs, to guide the process and maximize mental illness prevention and mental health promotion. Developing continuous and sustainable initiatives to promote the health and well-being of workers and organizations requires political, organizational and professional synergy in order to promote a multi-model intervention [24,32,33,40,43], preferably in a holistic and multidisciplinary perspective [18]. It is important to point out that “*a strong health workforce is a critical investment that contributes not only to improving the well-being of the population, but also to economic prosperity*” [43].

## Figures and Tables

**Figure 1 ijerph-19-05009-f001:**
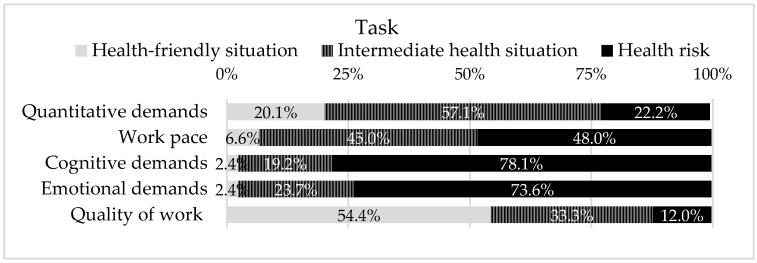
PRFs framed in the element Task of work system (of SEIPS 3.0 model), and the respective health risk that the exposure represents.

**Figure 2 ijerph-19-05009-f002:**
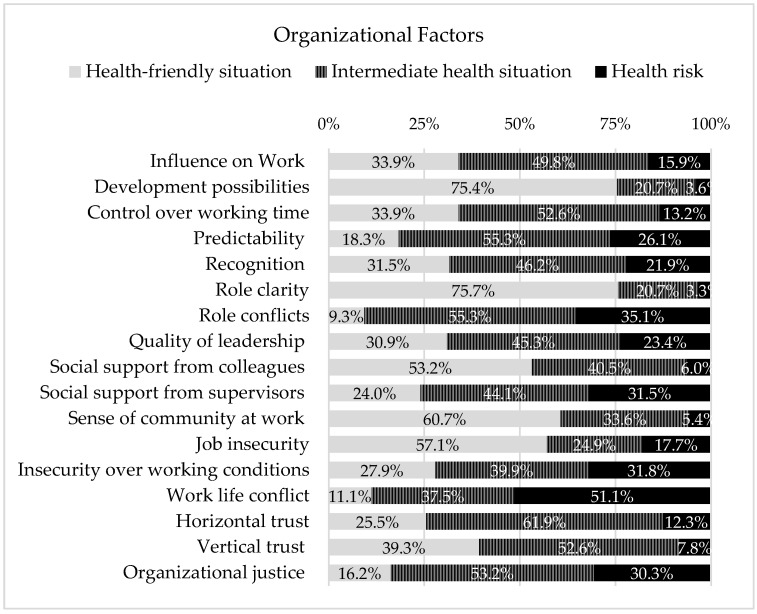
PRFs framed in the element of Organizational factors, of work system (of SEIPS 3.0 model), and the respective health risk that the exposure represents.

**Figure 3 ijerph-19-05009-f003:**
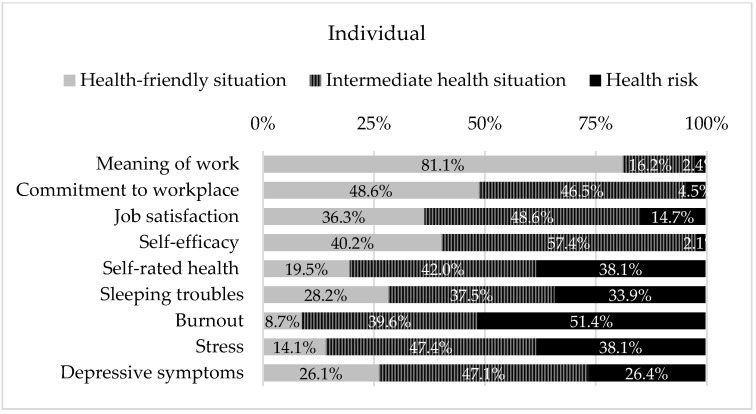
PRFs framed in the element of Individual, of work system of the SEIPS 3.0 model, and the respective health risk that the exposure represents.

## Data Availability

Data presented in this study are available on request from the corresponding author. The data are not publicly available due to privacy and ethical restrictions.

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
