# Peer review of "Healthcare Services and Formal Caregiver’s Psychosocial Risk Factors: An Observational Study"

_ijerph, 2022, doi:10.3390/ijerph19095009_

Round 1

Reviewer 1 Report

Comments are included in the attached file.

Author Response

Dear Miss Ioana Serban

Dear Guest Editor of the Special Issue

Submitted to section:“Global Health” https://www.mdpi.com/journal/ijerph/sections/public_health
Psychosocial Impact of Chronic Disease on Patients, Families and Caregivers
https://www.mdpi.com/journal/ijerph/special_issues/psychosocial_impact_chronic_disease_patients_families

 on the International Journal of Environmental Research and Public Health

Manuscript title: Healthcare Services and Formal Caregiver’s Psychosocial Risk Factors

Authors: Anabela Pereira *, Elisabeth Brito, Isabel Souto, Bruno Alves

Manuscript ID: ijerph-1595142

Thank you for your email dated 28 Feb 2022. We were pleased to know that our manuscript was again rated as potentially acceptable for publication in the International Journal of Environmental Research and Public Health, subject to adequate revision and response to the comments raised by the reviewers.

The manuscript was revised by modifying the Introduction, Materials and Methods, Results, Discussion, and Conclusions sections, based on the comments made by the reviewers.

As you notice, we agreed with almost all the comments raised by the reviewers. We would like to take this opportunity to express our sincere thanks to the reviewers who identified areas of our manuscript that needed corrections or modification and gave us the opportunity to improve the quality of the manuscript.

We appreciate your time and look forward to your response.

Reviewer 1

We appreciate the comments presented and the contribution to the improvement of the work. All comments were considered, and changes were made as proposed. More specifically:

  1. How the sample size was determinate:

Given the exploratory nature of the study, an a priori power analysis was not conducted. However, we have added information about the representativity of the sample, in this case, as a limitation, as well as relevant information about the population data and the desirability sample for future studies.

  1. It is recommendable to clarify this procedure and relocate it in the procedure section for data analysis. Additionally, how this cut-off points were determined (namely procedure adopted for COPSOQ scoring).

The rectification of this point was promptly accepted, and the content was relocated to the recommended section. In addition, it is clarified that the cut-off points adopted appear as one of the scoring strategy of COPSOQ II (Portuguese version), which is now clarified.

  1. This part of the research is not clear (namely content analysis) How the procedure was followed by psychologists to determinate the relationship between COPSOQ dimensions and the PRFs.

Effectively this section was unclear, so much so that it obviously conveyed the wrong idea (by our mistake). In this specific case, we tried to distribute the 31 PRFs (assessed through COPSOQ) across the 5 elements of the SEIPS 3.0 work system. Thus, the expert psychologists were guided by the original author's framework and description. As mentioned in the introduction, the 5 elements of SEIPS work system already comprise several PRFs, whose framework is described in the reference literature (e.g. Carayon & Smith, 2000). This point was also clarified.

  1. Clarification of objectives (no specific comment)

Was improved and clarified

Reviewer 2 Report

What is the main aim of the research?

It is not clearly defined what are the main results of this research?

The discussion and conclusions are not consistent enough. What are the concrete conclusions?  What is the professional and scientific contribution of this research in relation to the existing knowledge? The paper is written more as a review than a scientific one.

Author Response

Dear Miss Ioana Serban

Dear Guest Editor of the Special Issue

Submitted to section:“Global Health” https://www.mdpi.com/journal/ijerph/sections/public_health
Psychosocial Impact of Chronic Disease on Patients, Families and Caregivers
https://www.mdpi.com/journal/ijerph/special_issues/psychosocial_impact_chronic_disease_patients_families

 on the International Journal of Environmental Research and Public Health

Manuscript title: Healthcare Services and Formal Caregiver’s Psychosocial Risk Factors

Authors: Anabela Pereira *, Elisabeth Brito, Isabel Souto, Bruno Alves

Manuscript ID: ijerph-1595142

Thank you for your email dated 28 Feb 2022. We were pleased to know that our manuscript was again rated as potentially acceptable for publication in the International Journal of Environmental Research and Public Health, subject to adequate revision and response to the comments raised by the reviewers.

The manuscript was revised by modifying the Introduction, Materials and Methods, Results, Discussion, and Conclusions sections, based on the comments made by the reviewers.

As you notice, we agreed with almost all the comments raised by the reviewers. We would like to take this opportunity to express our sincere thanks to the reviewers who identified areas of our manuscript that needed corrections or modification and gave us the opportunity to improve the quality of the manuscript.

We appreciate your time and look forward to your response.

Reviewer 2

We appreciate the comments presented and the contribution to the improvement of the work. All comments were considered, and changes were made as proposed. More specifically:

  1. What is the main aim of the research?

Was improved.

  1. It is not clearly defined what are the main results of this research?

was improved and clarified

  1. The discussion and conclusions are not consistent enough. What are the concrete conclusions? What is the professional and scientific contribution of this research in relation to the existing knowledge? The paper is written more as a review than a scientific one

Several of the points raised have been clarified, with the recognition that this would be diluted throughout the discussion. The main results were summarized, in order to bring more clarity to the readers, as well as the scientific contributions. It should be noted that recent literature reviews point out that a limitation of the various intervention strategies for psychosocial risks in the work context is the fact that these interventions are not based on the specific needs of the various work contexts. Similarly, WHO documents reaffirm the need to adopt targeted strategies. For this reason, we believe that the dissemination of data aligned with these needs is, itself, a relevant contribution when we take an approach that goes beyond simple disclosure of data. That is, we to foresee these phenomena framed in its wider context, in this specific case, in its possible impact on the quality of health services, as well as in the journey of the patient.

Reviewer 3 Report

The conclusion must be more articulated to avoid that some statements are simply beautiful words or laudable wishes. Above all it is necessary that the Authors solve what they mean by "necessary to act also at the secondary and tertiary levels". Please remember that when the formal caregivers get sick they become patients themselves and benefit from the health system's treatments. Therefore the area in which Authors are concerned must be clearly defined: prevention or treatment of any NCDs caused by an unhealthy place and pace of work.

The juxtaposition of the words "continuous and sustainable" about suggested initiatives requires much more clarification: otherwise the word "sustainable" run the risk to be justified by reductive and limited choices due to economic needs; it is better for Authors to give examples.

Line 66, it should be better explained what the Authors mean by the term "personality" in order not to mislead Readers.

A more close link between the study presented and the NCDs appears to be lacking: missing it, the introduction appears less consistent than expected with the presentation of the study as well as discussion and conclusions.

Author Response

Dear Miss Ioana Serban

Dear Guest Editor of the Special Issue

Submitted to section:“Global Health” https://www.mdpi.com/journal/ijerph/sections/public_health
Psychosocial Impact of Chronic Disease on Patients, Families and Caregivers
https://www.mdpi.com/journal/ijerph/special_issues/psychosocial_impact_chronic_disease_patients_families

 on the International Journal of Environmental Research and Public Health

Manuscript title: Healthcare Services and Formal Caregiver’s Psychosocial Risk Factors

Authors: Anabela Pereira *, Elisabeth Brito, Isabel Souto, Bruno Alves

Manuscript ID: ijerph-1595142

Thank you for your email dated 28 Feb 2022. We were pleased to know that our manuscript was again rated as potentially acceptable for publication in the International Journal of Environmental Research and Public Health, subject to adequate revision and response to the comments raised by the reviewers.

The manuscript was revised by modifying the Introduction, Materials and Methods, Results, Discussion, and Conclusions sections, based on the comments made by the reviewers.

As you notice, we agreed with almost all the comments raised by the reviewers. We would like to take this opportunity to express our sincere thanks to the reviewers who identified areas of our manuscript that needed corrections or modification and gave us the opportunity to improve the quality of the manuscript.

We appreciate your time and look forward to your response.

Reviewer 3

We appreciate the comments presented and the contribution to the improvement of the work. All comments were considered, and changes were made as proposed. More specifically:

  1. The conclusion must be more articulated to avoid that some statements are simply beautiful words or laudable wishes. Above all it is necessary that the Authors solve what they mean by "necessary to act also at the secondary and tertiary levels". Please remember that when the formal caregivers get sick they become patients themselves and benefit from the health system's treatments. Therefore the area in which Authors are concerned must be clearly defined: prevention or treatment of any NCDs caused by an unhealthy place and pace of work.

We tried to improve the points, in order to bring more clarity to the readers. For the need for clarification on “prevention or treatment of any NCDs caused by an unhealthy work environment and pace”, we clarify It is not our purpose to propose any intervention for specific NCD's. We do not evaluate NCDs in healthcare professionals, even though we know (considering the literature) that exposure to PRF (and consequent stress at work), is closely associated with NCD’s development (we also clarify that point).

It is our intention to draw attention for the need to adopt measures to prevent PRFs, as they are of extreme relevance to the work system (according to SEIPs model), aiming at improving the quality of health services, and this has direct impact on their users, particularly patients with NCD's. As such, we clarify:

Considering the overall results, it realizes that prevention and intervention actions are urgent, in two ways: (1) to reply to health services caregivers’ considering mainly Task and Individual needs; (2) to improve the patient's journey through the health system (given relevance that these factors may have on the quality of health services as well as on their users)[4]. It is also important to mention that, given the relationship between exposure to work-related stress and the development of NCDs [17,41], prevention and intervention actions can also have a positive impact on the prevalence of NCD’s.

  1. The juxtaposition of the words "continuous and sustainable" about suggested initiatives requires much more clarification: otherwise the word "sustainable" run the risk to be justified by reductive and limited choices due to economic needs; it is better for Authors to give examples.

We appreciate the comment, and clarify that here “sustainable” is effectively intended to draw attention to the economic advantages associated with preventive action, which is why we add:

It is important to point out that “a strong health workforce is a critical investment that contributes not only to improving the well-being of the population, but also to economic prosperity”[43].

  1. Line 66, it should be better explained what the Authors mean by the term "personality" in order not to mislead Readers.

Although “personality” was adopted by the original author in the description of this element, we accepted the suggestion and changed it to “cognitive and psychosocial dimensions”.

  1. A more close link between the study presented and the NCDs appears to be lacking: missing it, the introduction appears less consistent than expected with the presentation of the study as well as discussion and conclusions.

We have presented some improvements, however, it is our intention to draw attention to the need to adopt measures to prevent PRFs, highlighted, given that they are extremely relevant to the work system (According to SEIPs), aiming at improving quality of health services, and this has a direct impact on their users, especially those with NCD's. We used the WHO's own recommendations in this sense, namely the WHO's focus on the need to improve health services in the face of NCD's through the empowerment of health professionals, as well as the SEIPS model that looks specifically at the journey of the patient with NCD's in the services of health

Reviewer 4 Report

The study is professionally performed and reported in most respects. I have only few (minor) points related to the presentation and discussion.

  1. While the innovative policy contribution of the study is clear and emphasized, less focus is on the innovative contribution to science and literature is not explicated. This should be briefly stated in a revision.
  2. In extension of 1., recommendations on how to proceed should be offered, i.e., what are the limitations and the scope of the present results, and where should future research take over.
  3. The sample is fairly limited in size, especially the number of private providers. It should e briefly discussed which impact(s) this may have on the outcome. Preferably, some simple power calculation should be added, so that recommendations for future studies sample size can be suggested. This would add to the value of the study, which might be questioned because of the small sample size.

Author Response

Dear Miss Ioana Serban

Dear Guest Editor of the Special Issue

Submitted to section:“Global Health” https://www.mdpi.com/journal/ijerph/sections/public_health
Psychosocial Impact of Chronic Disease on Patients, Families and Caregivers
https://www.mdpi.com/journal/ijerph/special_issues/psychosocial_impact_chronic_disease_patients_families

 on the International Journal of Environmental Research and Public Health

Manuscript title: Healthcare Services and Formal Caregiver’s Psychosocial Risk Factors

Authors: Anabela Pereira *, Elisabeth Brito, Isabel Souto, Bruno Alves

Manuscript ID: ijerph-1595142

Thank you for your email dated 28 Feb 2022. We were pleased to know that our manuscript was again rated as potentially acceptable for publication in the International Journal of Environmental Research and Public Health, subject to adequate revision and response to the comments raised by the reviewers.

The manuscript was revised by modifying the Introduction, Materials and Methods, Results, Discussion, and Conclusions sections, based on the comments made by the reviewers.

As you notice, we agreed with almost all the comments raised by the reviewers. We would like to take this opportunity to express our sincere thanks to the reviewers who identified areas of our manuscript that needed corrections or modification and gave us the opportunity to improve the quality of the manuscript.

We appreciate your time and look forward to your response.

Reviewer 4

We appreciate the comments presented and the contribution to the improvement of the work. All comments were considered, and changes were made as proposed. More specifically:

  1. While the innovative policy contribution of the study is clear and emphasized, less focus is on the innovative contribution to science and literature is not explicated. This should be briefly stated in a revision.

was improved.

  1. In extension of 1., recommendations on how to proceed should be offered, i.e., what are the limitations and the scope of the present results, and where should future research take over.

was improved.

  1. The sample is fairly limited in size, especially the number of private providers. It should e briefly discussed which impact(s) this may have on the outcome. Preferably, some simple power calculation should be added, so that recommendations for future studies sample size can be suggested. This would add to the value of the study, which might be questioned because of the small sample size.

Given the exploratory nature of the study, an a priori power analysis was not conducted. However, we have added information about the representativity of the sample, in this case, as a limitation, as well as relevant information about the population data and the desirability sample for future studies.

Round 2

Reviewer 2 Report

A corrected and improved version of this paper in my opinion can be published. 

Author Response

Thank you!